# The Simulation Game—Virtual Reality Therapy for the Treatment of Social Anxiety Disorder: A Systematic Review

**DOI:** 10.3390/ijerph182413209

**Published:** 2021-12-15

**Authors:** Pasquale Caponnetto, Sergio Triscari, Marilena Maglia, Maria C. Quattropani

**Affiliations:** 1Department of Educational Sciences, Section of Psychology, University of Catania, 95124 Catania, Italy; sergio.triscari@studium.unict.it (S.T.); maria.quattropani@unict.it (M.C.Q.); 2Center of Excellence for the Acceleration of Harm Reduction (COEHAR), University of Catania, 95123 Catania, Italy; 3CTA-Villa Chiara Psychiatric Rehabilitation Clinic and Research, 95030 Mascalucia, Italy

**Keywords:** virtual reality, virtual reality exposure therapy, social anxiety disorder, social phobia, VRET, cyberpsychology

## Abstract

*(1) Background:* With the term Virtual reality (VR) we refer to a three-dimensional environment generated by the computer, in which subjects interact with the environment as if they were really inside it. The most used VR tools are the so-called HMD (head-mounted display) which make it possible to achieve what theorists define “direct mediated action”. The aim of our systematic review is specifically to investigate the applications of virtual reality therapy for the treatment of social anxiety disorder, also known as social phobia. The most common treatment for social anxiety disorder is represented by “in vivo exposure therapy” (iVET). This method consists of exposing the participant, in a gradual and controlled way, to anxious stimuli, with the goal to change the subject’s response to the object or situation that is causing the fear. However, the main flaw of “in Vivo therapies” is represented by both the huge costs involved and the possible disturbance variables that can hinder the execution of the therapeutic treatment. Virtual reality exposure therapy could therefore, if confirmed in its effectiveness, constitute a solution to eliminate these two defects demonstrated by “in vivo exposure therapy”. The goal is to use VR as a means for the clinician to build a tailor-made path for the participant in order to make him acquire “in virtual” those skills necessary for a good adaptation in the “real” world. *(2) Methods:* From February 2021 until the date of submission of the article (September 2021), we conducted a systematic review aiming to verify the effectiveness of virtual reality exposure therapy (VRET) for the treatment of SAD. *(3) Results:* We identified a total of 205 unique articles. Among these, 20 full-text articles were assessed for eligibility and 5 of these met the eligibility criteria and were, therefore, included in the final systematic review. *(4) Conclusions:* Virtual reality therapies proved to be a valid alternative to the acquisition of social skills suitable for improving the symptoms of SAD. Although there has not been a significant difference between VRET and iVET, the low costs and flexibility of VRET open up new scenarios for achieving greater psychophysical well-being.

## 1. Introduction

With the term Virtual Reality (VR) we refer to a three-dimensional environment generated by the computer, in which subjects interact with the environment as if they were really inside it [1]. Virtual reality represents a turning point in the human–computer relationship, as it is able to make the user experience the “sense of presence”, thus making the fundamental transition from the sensation of “perceiving information” to the sensation of “being in the place of information” [2].

There are several incremental levels of simulation provided by Virtual Reality. In particular, a distinction is made between: “Augmented Reality” (AR) when it is possible to superimpose computer-generated images on reality; “Non Immersive Virtual Reality” (Desktop VR) when, for example, we are faced with devices equipped with stereoscopic 3D such as modern televisions; “Immersive Virtual Reality” (IVR) when all the perceptual channels of the subject are isolated and “total” immersion is experienced.

In light of this, the IVR built and digitally manipulated represents the “best” level to carry out a direct mediated action. The subject thus becomes an active creator of his experience, thanks to an immersive technology that not only gives him the feeling of being physically present in the virtual world that surrounds him, but, above all, allows him to interact with it [3].

Today, the most used immersive virtual reality devices are represented by head-mounted displays, often accompanied by joysticks or Data Gloves—even if complete haptic suits are being developed, capable of further redefining the boundaries of “sense of presence”. The reason why we feel so present within the environment built by VR is because virtual reality employs simulation mechanisms very close to those used by our mind [4]. In essence, we can say that our mental system is itself a simulation system of reality. The confirmation of this is given by the innovative discovery of Giacomo Rizzolatti [5] and collaborators, who identified the existence of two groups of bimodal visuo-motor neurons, namely “canonical” neurons and “mirror” neurons that confirm the existence of a simulation system in our mind. In the clinical setting, VR systems have shown that they can represent a credible, realistic and effective perspective, as well as easily adaptable to different psychotherapeutic approaches [6].

The opportunities offered by VR systems to the field of experimental psychology are numerous: first of all, we can add to the X and Y coordinates, the Z coordinate, or the depth—this makes, together with the possibility of active interaction from the participant, the perception of virtual space similar to the perception of real space. Secondly, we can completely control the possible disturbance variables that intervene in a negative way during the treatment.

The goal is to use VR as a means for the clinician to build a tailor-made path for the participant in order to make him acquire “in virtual” those skills necessary for a good adaptation in the “real” world. In fact, by providing users with a highly realistic, flexible, engaging, safe and controllable simulation, they are able to acquire the skill, confidence, mental and psychophysical preparation to face real-world activities [7].

The fields of psychological application in which this is possible are manifold, from phobic disorders and PTSD (Post Traumatic Stress Disorder) to autism, attention deficit hyperactivity disorder (ADHD), eating disorders (ADD), panic (DAP), schizophrenia, and neuropsychological rehabilitation.

The aim of our systematic review work is specifically to investigate the applications of virtual reality therapy for the treatment of social anxiety disorder, also known as social phobia. This disorder falls into the DSM-V category of “Anxiety Disorders” and is characterized by an “excessive and irrational fear of the social situations in which the individual is exposed”. Analyzing the literature, we found many systematic reviews that deal with anxiety disorders and phobias (Wechsler et al. [8]; Freitas et al. [9]; Kelson et al. [10]; Krzystanek et al. [11]). However, no one focuses exclusively or deeply enough on Social Anxiety Disorder per se.

The complexity of this disorder, therefore, requires a review exclusively dedicated to it in order to prepare the ground for future experimental studies with technologies ever closer to “reality”.

In particular, the declinations of social anxiety disorder, on which we focused, can be identified as: performance/exam anxiety, public speaking anxiety, difficulty in dealing with situations in which the individual is at the center of the attention. Social phobia, in fact, is a rather widespread disorder among the world population—according to some studies, the percentage of people who suffer from it varies from 3% to 13%.

The most common treatment for social anxiety disorder is represented by “in vivo exposure therapy” (iVET). This method consists of exposing the participant, in a gradual and controlled way, to anxious stimuli, in order to change the participant’s response towards the object or situation that is causing the fear. However, the main flaw of “in vivo therapies” is represented by both the huge costs involved and the possible disturbance variables that can hinder the execution of the therapeutic treatment. Virtual reality exposure therapy could, therefore, if confirmed in its effectiveness, constitute a solution to eliminate these two defects demonstrated by “in vivo exposure therapy”.

In this regard, the cost sustainability for the VR intervention was analyzed by Robillard et al. [12], who validate the SWEAT questionnaire, which measures the costs and effort required to conduct exposure in vivo or in VR. In their research, after the evaluation of 265 exposure sessions (in vivo = 140; in virtuo = 125) it was shown that conducting VR exposure is less expensive and more easily adaptable to the needs of patients.

VR technology systems allow the infinite replicability of the anxious stimulus and the modularity of the difficulty levels of the interactions.

In fact, if in the classic “In vivo exposure therapy” for the treatment of SAD the presence and availability of a more or less varied clinical research team is necessary, in the “Virtual Reality Exposure Therapy”; instead, everything can be performed digitally and without particular time limits.

The cost of maintaining such a large team in in Vivo therapy, as well as the cost of time to perform the procedures, is consequently halved as the team itself can be reduced to a few doctors responsible for managing the therapy and technology, with return on the price to pay for the patient.

Several criticisms have been advanced with respect to this perspective. For many authors, even VRET has an inherent management cost; however, as Giuseppe Riva [4] states, virtual reality has grown rapidly in the last decade and costs have decreased. One should look at how the top-of-the-range HMD currently on the market cost a few hundred dollars, whereas rehabilitation programs require licenses that are not extremely expensive.

With this in mind, VRET (Virtual Reality Exposure Therapy) can be used as a support tool for psychotherapy to improve the quality of life of this population. There are many socio-cultural implications that an innovative therapy such as VRET, if confirmed in its effectiveness, can offer to those suffering from social anxiety disorder. We will, therefore, provide an updated review of VR therapeutic techniques and their effectiveness in clinical practice in order to reduce or defeat this disabling disorder, and finally we will try to understand if VRET produces better results than iVET.

## 2. Materials and Methods

### 2.1. Research Object

The purpose of this research is to verify the effectiveness of virtual reality therapy for the treatment of Social Anxiety Disorder (or Social Phobia) with particular reference to its forms of: examination/interview/performance anxiety, anxiety in public speaking, and anxiety in dealing with situations in which the subject is the center of attention.

### 2.2. Search Strategy

The systematic review was completely carried out according to the PRISMA 2020 guidelines for systematic review by PRISMA Group [13]. The bibliographic research was carried out from February 2021 until the date of submission of the article (September 2021) in the databases of the PubMed, Psycnet, ResearchGate sites using the following string of search terms: “Virtual Reality (or “VR”) and Social Anxiety Disorder”; “Virtual reality (or “VR”) and social phobia”; “Virtual reality (or “VR”) and anxiety test”; “Virtual Reality (or “VR”) and Public Speaking”; “Virtual reality (or “VR”) and interview anxiety”; “Virtual reality (or “VR”) and performance anxiety”.

### 2.3. Eligibility Criteria

We have included every article written in English with no time limit on publication date, meeting the following criteria:(1)***Participants***: Patients diagnosed with social anxiety disorder and without further diagnosis of mental illness.(2)***Intervention***: Immersive Virtual Reality Therapy using a head-mounted display (HMD) and a digitally recreated virtual environment.(3)***Comparison***: symptoms before “VRET (Virtual Reality Exposure Therapy) with immersive virtual reality technologies” and “post-treatment” symptoms.(4)***Outcome***: we considered the post-treatment symptoms related to the disorder, whether or not there was the acquisition of social skills, and whether or not there was a greater adaptation to the social environment.

### 2.4. Data Extraction

The data was extracted using a format that included for each article: author, year, title, nation where the research took place, type of study, sample, measures used, results, and follow-up if any.

### 2.5. Risk of Bias Assessment

The risk of bias for the included studies was assessed with Cochrane risk-of-bias tool for randomized trials, version 2 (RoB 2) by Sterne et al. [14,15].

## 3. Results

### 3.1. Search Results

The articles resulting from the search phase in the databases listed above produced a total of 386 articles. Another noteworthy article was also identified during the review that was not detected from the research but that many other analyzed articles cited. After this first search, 182 duplicates were eliminated, thus identifying 205 unique articles. The small number of studies confirms that this is a new field and that its potential has still to be fully explored.

These articles were subject to further scrutiny, through the analysis of the titles and abstracts, which led to the elimination of an additional 185 articles, not compatible with the theme of the paper. The remaining 20 articles were read in their entirety, 15 of which were excluded because they were not meeting the eligibility criteria. Finally, the number of studies included in the qualitative summary was 5.

The above description is summarized in the flowchart in Figure 1, whereas the data extraction of these studies can be viewed in Table 1. Finally, the control analysis with the RoB 2 was summarized in Figure 2.

### 3.2. Characteristics of the Included Studies

The research by Anderson et al. [16] represents one of the first randomized studies that make a comparison between virtual reality exposure therapy and “in Vivo” exposure therapy with regard to social anxiety disorder. In total, 97 participants, with an average age of 39 years, predominantly women and meeting the criteria for social anxiety disorder, verified through the “Structured Clinical Interview” (SCID) for the DSM-IV, were randomly assigned to three groups: VRET (Virtual Reality Exposure Therapy); EGT (Exposure Group Therapy); control group on the waiting list.

These participants identified the declination of “public speaking” as their primary social fear. Specifically, the measures used in the research were: the “Personal Report of Confidence as a Speaker” (PRCS); the “Fear of Negative Evaluation—Brief Form” (FNE-B); the “Behavioral Avoidance Test”; the Clinician Global Impressions of Improvement (CGI); and the “Credibility and Expectancy Questionnaire” (CEQ).

Participants completed all self-assessments at pre-treatment, post-treatment and follow-up of 3 and 12 months—while the diagnostic evaluations were drawn up by doctoral candidates who were blind to the type of treatment. Both treatments were administered according to a manualized protocol for eight sessions. The VRET and EGT treatment groups were designed to be as similar as possible, with the exception of how exposure was delivered. Specifically, as far as VRET is concerned, the participants used an HMD that introduced them into a virtual environment, built by the experimenters, such as an auditorium where the therapist could manipulate the reactions of the audience while the subject was intent on pronouncing a speech.

Regarding the EGT, it was conducted with the same “task” within an “In Vivo” group, consisting of a maximum of five collaborators and the real participant who delivered the speech. From the data analysis, we can observe how the two active treatments (VRET and EGT) showed a similar improvement in most of the measures. Participants in fact reported high expectations for a positive result after the first treatment session and satisfaction at the end of the sessions.

The results of this randomized clinical trial demonstrate that VRET is effective in reducing fears of public speaking among those diagnosed with social anxiety disorder—improvement is also maintained 1 year later.

The research by Bouchard et al. [17] represents the second randomized study analyzed, which compares VRET (virtual reality exposure therapy) and “In Vivo” therapy for social anxiety disorder in the context of a cognitive behavioral psychotherapy. In line with the other research covered in our systematic review, to ascertain the presence of a social anxiety disorder, the participants were interviewed through the Structured Clinical Interview for DSM-IV (SCID).

All diagnoses were reviewed and confirmed by a second evaluator to increase reliability—in particular, within the research a primary diagnosis of SAD was required for at least the last 2 years. All diagnoses then underwent a further review that also met the DSM-V diagnostic criteria. Participants (*n* = 59) were randomly assigned to three different conditions: CBT in VRET (*n* = 17); In Vivo CBT (*n* = 22); and Control group on the waiting list (*n* = 20). The exhibition “in Vivo” consisted of role-playing games, inside and outside the therapist’s office: in which the participant, for example, formulated bizarre requests at shops or delivered an embarrassing speech in front of collaborators. In the VR exhibition, instead, participants used HMDs and motion detectors, experimenting with different scenarios, created digitally by the experimenters, such as the simulation of a job interview or a speech in front of the public (It is also important to underline the presence of a neutral scenario for the first session, which is useful for becoming familiar with the virtual environment). The treatment therefore consisted of 14 weekly therapy sessions (in vivo or in virtual), of 60 min each. The objective of the exhibition was to develop a new, adaptive and non-threatening vision of the feared social situations, and to verify the effectiveness of the use of virtual reality on this objective.

Regarding the clinical outcomes of the research, the authors used self-administered assessments: both before and after treatment for each group, and during the 6-month follow-up, conducted with the CBT groups. The main outcome was identified by the authors as the “total score” of the Liebowitz Social Anxiety Scale-Self Reported (LSAS-SR) which evaluates fear and avoidance of a series of social interactions and performance situations.

Other scales used were: the Social Phobia Scale (SPS), the Social Interaction Anxiety Scale (SIAS), the Fear of Negative Evaluation (FNE), and the Beck Depression Inventory (BDI-II) to measure potential depressive symptoms associated with the disorder. The researchers also asked the participants of the various groups to carry out a Behavioral Assessment Task (BAT), both before the first therapy session and after the last session.

Specifically, patients had to deliver an impromptu speech with the instruction to make it last as long as possible (for a maximum of 6 min). This videotaped speech was then evaluated by three independent blinded evaluators using the Social Performance Rating Scale (SPRS). The research also includes measures relating to resources, advantages and difficulties encountered with exposure therapies, and in particular with VRET: In order to study the practical and financial resources necessary for the sessions, in fact, therapists fill out the SWEAT questionnaire (Specific Work for Exposure Applied in Therapy) after each therapy session—the elements of the SWEAT actually measure topics such as effort in terms of the cost, time, and planning required to develop and conduct the exhibit and the difficulties encountered (such as problems with hardware). On the other hand, with the Simulator Sickness Questionnaire (SSQ) the side effects induced by HMD, commonly known as Cybersickness, are measured.

Finally, the central element of VRET, namely the “Sense of Presence” is measured through the Presence Questionnaire (PQ) and the Gatineau Presence Questionnaire (GPQ). From the analysis of the data, statistically significant results emerge in favor of exposure therapies. Furthermore, CBT with VRET proves to be slightly more effective than CBT in Vivo with regard to LSAS-SR and SPS. All benefits were then maintained at the 6-month follow-up.

The research by Kampmann et al. [18] represents, in our review, the third randomized controlled study on the efficacy of VRET, applied to participants with social anxiety disorder. The sample of the study was composed in particular of 60 subjects who met the criteria for the diagnosis of SAD, verified through the Structured Clinical Interview (SCID) for the DSM-IV. Participants were randomly assigned to one of three conditions: VRET (Virtual Reality Exposure Therapy); iVET (in Vivo Exposure Therapy); control group on the waiting list. After obtaining informed consent, eligible participants underwent a pre-test and a post-test, including a series of measures: The main outcome was the comparison of “pre” and “post” treatment social anxiety symptoms, measured with the Liebowitz Social Anxiety Scale-Self Report (LSAS-SR)—and the “subjective fear of being negatively evaluated by others in social situations” measured with the Fear of Negative Evaluation Scale-Brief (FNE-B). Other measures used were: the Depression Anxiety Stress Scales (DASS-21), Personality Disorder Belief Questionnaire (PDBQ), the Eurohis Quality of Life Scale (EUROHIS-QOL)—also administered “pre” and “post” treatment.

Finally, a behavioral assessment task was proposed in which speech duration and performance were assessed.

A stopwatch was used to assess duration, whereas to assess speech performance, two independent judges, blind by condition and rating point, rated the videotaped speeches using 17 elements of a public speaking performance measure on a 5-point Likert scale by Rapee and Lim, 1992 [19]. A follow-up was performed after 3 months. VRET took place in a laboratory of the University of Amsterdam. Participants were made to use HMDs, through which they could interact with computer-generated situations by the experimenters.

The virtual situations covered one-to-one and group situations, aimed at causing anxiety in individuals with SAD, such as: giving a speech in front of an audience asking questions, talking to a stranger, buying and returning clothes, participating in a job interview, being interviewed by reporters, having dinner in a restaurant with a friend, and having a blind date.

Semi-structured therapist-controlled dialogues ensured a certain level of interaction between the virtual characters and the participant. Similarly, iVET consisted of gradual exposure therapy to real-life situations. From the data analysis, it emerges that the two treatment conditions, VRET and iVET, correlate positively with a better assessment of social anxiety, avoidance, and perceived stress.

Contrary to what was expected, iVET proved superior to VRET in terms of the decrease in SAD symptoms and the increase in speech performance. Furthermore, although VRET actually reduced anxiety and avoidance in social situations, it did not significantly reduce the fear of negative evaluation, which is a key cognitive feature of SAD. Despite this, the overall results indicate that VRET has the potential to produce positive and generalizable effects to real social situations. However, the reason why iVET was superior to VRET may find an answer within the historical context of the research.

In 2015/2016, the HMDs were still “immature” and probably did not allow the same level of presence as an in vivo therapy, which is why the importance of working on photorealism and interactivity of virtual environments is emphasized. The fourth study we dealt with is the research by Kim et al. [20], which tried to analyze the effectiveness of a participatory and interactive VR intervention on SAD.

In total, 32 participants “with SAD” and 33 “healthy” participants—with an average age of 23 years and without significant differences in terms of gender—completed the study. Participants with SAD met the DSM-IV diagnostic criteria.

The study was part of a larger project conducted to evaluate the effects of interactive and participatory VR solutions across psychological scales, functional near-infrared spectroscopy, functional magnetic resonance, and different physiological signals. Among these, however, only the results of the psychological scale were analyzed for this study in order to center the topic. The VR VIVE viewer (owned by the HTC company) was used for the intervention. The VR intervention was designed to allow participants to perform a total of six sessions. Specifically, participants were allowed to run two sessions in a row in a single visit, and the first session was started at an easy level. During the second session, however, participants could select the desired level. The possibility of stopping at any time during the VR experience was also explained to them, and that the researchers would be present throughout the experience to deal with any unexpected events.

In the virtual situation, between 7 and 8 non-player characters appeared and presented themselves and listened to an introductory speech given by the participant. As the difficulty level increased, the attitudes of non-player characters who listened to the participant’s introduction changed as they became distracted and chatted with each other. On the hard level, one of the non-player characters challenged the participant while he was introducing himself by saying, “Please introduce yourself correctly”.

To measure the effectiveness of the treatment, the participants completed a battery of assessments to assess their psychological state before and after the therapeutic sessions. The measures used were: the “Beck Anxiety Inventory” (BAI); the “State-Trait Anxiety Inventory” (STAI); the “Social Phobia Scale” (SPS); the “Social Interaction Anxiety Scale” (SIAS); the “Brief-Fear of Negative Evaluation Scale” (FNE-B); the “Internalized Shame Scale” (ISS); the “Post-Event Rumination Scale” (PERS); and the “Liebowitz Social Anxiety Scale” (LSAS). General anxiety symptoms as measured by BAI and STAI-T significantly improved after treatment, whereas STAI-S did not improve significantly.

SPS, SIAS, KSAD, BFNE, and LSAS, which are measures for assessing symptoms of SAD, all showed significant improvement after VR. The ISS showed significant improvement on the overall scale and on the subscales of emptiness, self-punishment and fear of error. There was no significant difference in inadequacy. Negative rumination, which is a subscale of PERS, showed significant improvement after VR treatment, but positive rumination did not.

After the six overall sessions, it was analyzed whether the scores on the psychological scales differed in the SAD group compared with those in the healthy group. The results showed that even after completing the VR sessions, the SAD group continued to score significantly higher than the healthy control group on all psychological scales. In other words, although significant improvements in SAD symptoms were found after treatment, these symptoms were still significantly different than those in the healthy group.

This suggests that the effect of virtual reality treatment alone may not be sufficient to achieve the subject’s response or remission. In any case, the results must be evaluated in the context of the research, which had both the limit of not having a real control group on the waiting list and the limit of using self-assessment scales (a limit present, in part, also in the other research) that could be “disturbed” by negative self-perceptions of participants regardless of their performance and their real status.

Finally, we examine the study by Wallach et al. [21]. This study is chronologically placed before all the others and for this reason it can represent a “precursor” of the studies on virtual reality since in 2009 the technology was not very widespread. The aim of the study was to verify the effectiveness of Virtual Reality associated with cognitive-behavioral therapy for the treatment of Social Phobia, and in particular in its declination of “Public Speaking Anxiety”.

Specifically, 88 participants were randomly sorted into the three conditions: VRET with CBT (*n* = 28); CBT in Vivo (*n* = 30); control group on the waiting list (*n* = 30). The treatment, specifically, consisted of role-playing games, to be performed in “reality” or “virtual”, the latter through HMD that generated situations on the computer in which the participant read a text in front of an audience that applauded, asked questions, or was hostile. Each participant was therefore administered the Liebowitz Social Anxiety scale (LSAS); self-statements during public speaking (SSPS); and the fear of negative evaluation (FNE). Finally, a “Behavioral Task” was proposed in which at the end of the treatment the participants conducted 10 min of conversation on a topic of their choice in front of a live audience, and were evaluated on 10 anxiety indicators.

The results once again show a significant positive correlation between exposure therapies (both CBT and VRCBT) and the improvement of various measures with the exception of FNE—results also confirmed at one-year follow-up, published separately [22]. The use of Virtual Reality also seems to overcome many difficulties related to classical therapy, such as costs and responsiveness of the subjects, and, therefore, represents an attractive alternative therapy.

What is surprising in this study is that already in 2009, with not yet advanced technologies, the subjects responded positively to VRET, demonstrating how working in this field is really profitable for achieving better psychophysiological well-being.

## 4. Discussion

So far, this systematic review represents one of the first attempts to systematically examine the studies that have compared the efficacy of Virtual Reality Exposure Therapy (VRET) with In Vivo Exposure Therapy (iVET) for Social Anxiety Disorder. Confirming our initial hypotheses, VRET has proved to be a valid alternative to “In Vivo” therapies for the treatment of Social Anxiety Disorder and its various forms. From the various analyzed research, it emerges that this therapy produces significantly positive results in a range that goes from 6 to 14 sessions. The research analyzed is positively correlated with a better diagnosis of the main measure of SAD. Moreover, the studies that contemplate a follow-up show that the curve of improvement is maintained over time. By critically evaluating the research we observe how in the research by Anderson et al. [16], PRCS and BAT measures show a significant effect of active therapies compared with the waiting list control group. There are no significant differences between EGT and VRET except for FNE-B, which only improves for EGT. In the research by Bouchard et al. [17], the results were found to be consistent with other research. At post-treatment, VR was more effective than traditional exposure on the primary outcome measure (LSAS-SR) and on one of the five secondary outcome measures (SPS), whereas it was slightly less effective on the FNE measure. The result on SWEAT also gives us empirical confirmation of our hypothesis that VRET would be a simpler and cheaper intervention than iVET at SAD, thereby allowing the possibility to offer more exposure experiences. Bouchard et al. [17] highlighted the importance of the therapeutic alliance in predicting the outcome of SAD symptoms. In the study by Kampmann et al. [18], for example, where iVET was in some cases (FNE-B; EUROHIS-QOL) superior to VRET, the participant and the therapist were in two separate rooms during exposure to virtual reality. The absence of therapist support may have negatively impacted the therapeutic alliance, and thus may have reduced the effectiveness of VRET. Still in the research of Kampmann et al. [18], the regression analysis demonstrated the efficacy of VRET and IVET at post-treatment for LSAS-SR, BAT, PDBQ, DASS-21 measures. However, there are still many limitations: in the research of Kim et al. [20], although nearly all measures improved with VRET treatment (LSAS-SR; BAI; STAI; SPS; SIAS; PERS; ISS; FNE-B) the latter was unable to carry participants with SAD at the level of “healthy” participants. Another limitation of the studies is the frequent use of self-assessment measures which may not reflect the real levels reached. Most of the research (Anderson et al. [16]; Kampmann et al. [18]; Wallach et al. [21]) shows that the FNE-B measurement achieves positive results only through “In Vivo” therapy. This result can be interpreted in the perspective of a “realism” not yet achieved by available technologies, which does not allow participants to completely reduce their social anxiety. Much remains to be done to improve the technology behind VR exposure and thereby, the efficacy of VRET. However, by analyzing the research chronologically, in the various measures in common, we can still confirm a gradual improvement, in line with technological development, of the exposure in virtuo. The hope is, therefore, the achievement of an even more “mature” technology that can make a difference in the treatment of this debilitating disorder.

Put another way, the superiority of VRET over iVET should not be seen as much in the perspective of the reduction in symptoms, since they seem to be equally effective, but in the drastic reduction in the costs to carry out the therapy and in the flexibility that allows the clinician to control all the variables at stake. The low cost of VRET, in fact, may today represent the turning point for a broader access to psychological care to socioeconomic classes that are currently excluded.

In addition, Virtual Reality’s flexibility opens up new psychotherapist scenarios in which the risks that a “disturbance variable” could compromise the therapeutic work are eliminated. Worth nothing, the analyzed research was exclusively based on Cognitive Behavioral Therapy, thus it would be interesting to hypothesize the support of Virtual Reality with other psychotherapeutic approaches.

Virtual Reality is not free from limits, among which the main one is represented by so-called “cybersickness”. The hope in this regard is the development in the following years of hardware and software technologies that can reduce this feeling of nausea and allow for an even longer “exposures”. Of course, VR therapy is a tool that does not replace the founding elements of the therapeutic relationship: dialogue and listening between therapist and patient. Rather it has to be seen as an integrated approaches to the clinical practice in which the therapist keeps nurturing the human contact with the patient by creating a dialogue between classic psychotherapy and new technology. Of particular interest is exploring the therapeutic process insofar it is related to the outcome, and it is paramount to understanding mechanisms of change during therapy.

However, future research in this area should evaluate the effects of virtual reality exposure in an even longer term. It should also always include a measure of the “sense of presence” as this is what makes virtual reality a “transformative reality for the subject” [23]. In conclusion, standard data collection protocols should be improved in order to overcome self-assessment measures and generate more accurate measures.

The study has the following limitations: first of all, the low number of included studies then the protocols employed in the above-described scientific papers present potential limitations such as low number of participants/example data, low variability in the data collection, and no comparison with respect to other methods.

## 5. Conclusions

In sum, virtual reality treatments seem to be an applicable option for decreasing the symptoms of SAD through the social skills learning. Somewhat surprisingly, as highlighted by our results, the efficacy of VRET is tantamount to iVET. Nonetheless, because of the small number of studies included in this systematic review, many important questions are left unanswered, such as the repercussions for therapeutic alliance or the use of other instruments than the self-report measures, for example, that should be addressed in future research.

The future of Virtual Reality treatments is currently promising and will face new challenges in the coming years. There is a general need to understand how new technologies, given their transformative potential, can find a place within the therapeutic practice [24].

## Figures and Tables

**Figure 1 ijerph-18-13209-f001:**
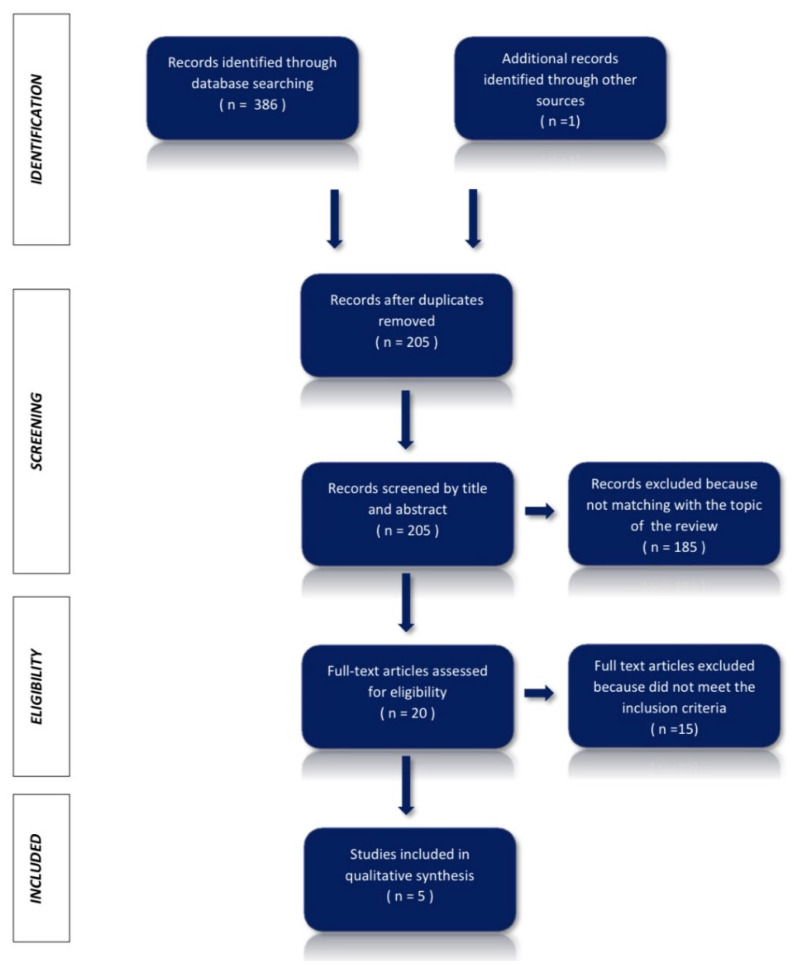
PRISMA (2020)—Flow Diagram.

**Figure 2 ijerph-18-13209-f002:**
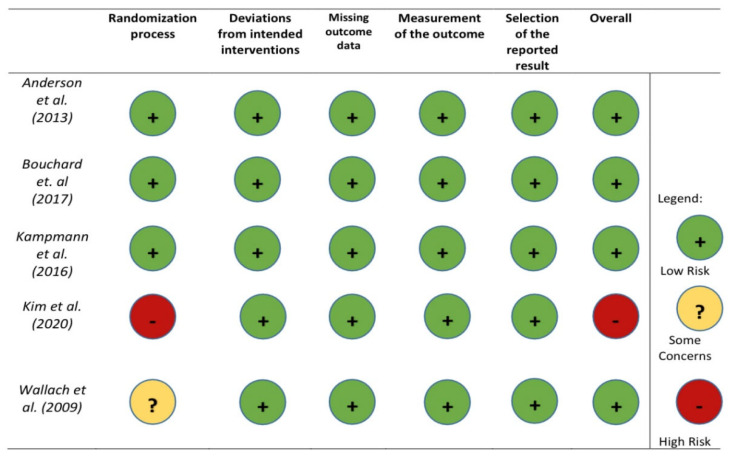
Cochrane risk-of-bias tool (RoB 2).

**Table 1 ijerph-18-13209-t001:** Data Extraction. Authors included in the table in alphabetical order starting from the first surname.

Authors	Year	Title	Nation	Type of Study	Sample	Measures	Follow Up	Results
*Page L. Anderson;* *Matthew Price;* *Shannan M. Edwards; Mayowa A. Obasaju;* *Stefan K. Schmertz;* *Elana Zimand;* *Martha R. Calamaras*	2013	Virtual reality Exposure therapy for Social Anxiety Disorder: A randomized Controlled Trial	USA	Randomized Controlled Trial	97	SCID PRCS FNE-B BAT CGI CEQ	YES (3 Months) (1 Year)	The two active treatments (VRET-EGT) have showed similar improvement on most measures. The improvement is also maintained for up to 1 year.
*Stéphane Bouchard;* *Stéphanie Dumoulin;* *Geneviève Robillard;* *Tanya Guitard;* *Évelyne Klinger;* *Hélène Forget;* *Claudie Loranger;* *François Xavier Roucaut*	2017	Virtual Reality compared with in vivo exposure in the treatment of social anxiety disorder: A three-arm randomised controlled trial	Canada	Randomized Controlled Trial	59	SCID LSAS-SR SPS SIAS FNE BDI-II BAT SPRS SWEAT SSQ PQ GPQ	YES (6 Months)	Both VRET and “in vivo” CBT were statistically significant for improving SAD-related measures. Specifically, VRET has been shown to be more effective than “in vivo” therapy for LSAS-SR and SPS. All benefits were maintained at the 6-month follow-up.
*Isabel L. Kampmann;* *Paul M. G. Emmelkamp;* *Dwi Hartanto;* *Willem-Paul Brinkman;* *Bonne J. H. Zijlstra;* *Nexhmedin Morina;*	2016	Exposure to virtual social interactions in the treatment of social anxiety disorder: A randomized controlled trial	The Netherlands	Randomized Controlled Trial	60	SCID LSAS-SR FNE-B DASS-21 PDBQ EUROHIS-QOL PDBQ BAT	YES (3 Months)	The two treatment conditions (VRET and iVET) correlate positively with a better assessment of social anxiety, perceived stress and avoidance. Contrary to what iVET was expecting proved to be superior to VRET due to the decrease in SAD symptoms and increased speech performance.
*Hyun-Jin Kim;* *Seulki Lee;* *Dooyoung Jung;* *Ji-Won Hur;* *Heon-Jeong Lee;* *Sungkil Lee;* *Gerard J. Kim;* *Chung-Yean Cho;* *Seungmoon Choi;* *Seung-Moo Lee;* *Chul-Hyun Cho;*	2020	Effectiveness of a Participatory and Interactive Virtual Reality Intervention in Patients With Social Anxiety Disorder: Longitudinal Questionnaire Study	Korea	Longitudinal Study	65	M.I.N.I. KSAD BAI STAI SPS SIAS FNE-B ISS PERS LSAS-SR		All measures improved after VRET treatment. Despite this, the intervention in VR was not sufficient to bring the subjects with SAD to the level of healthy subjects. In this perspective, one of the limits of the study compared with the others it is represented by the lack of a real control group on the waiting list.
*Helene S. Wallach;* *Marilyn P. Safir;* *Margalit Bar-Zvi;*	2009	Virtual Reality Cognitive Behavior Therapy for Public Speaking Anxiety A Randomized Clinical Trial	Israel	Randomized Controlled Trial	88	LSAS SSPS FNE BAT	YES (1 Year)	A significant correlation emerges between exposure therapies and improvement in anxiety levels. The FNE measurement does not have a significant improvement.

**Legend**: **BAI** = Beck Anxiety Inventory; **BAT** (Anderson et.al.) = Behavioral Avoidance Test; **BAT** (Bouchard et al.|Kampmann et al.|Wallach et al.) = Behavioural Assessment Task; **BDI-II** = Beck Depression Inventory; **CEQ** = Credibility and Expectancy Questionnaire; **CGI** = Clinician Global Impressions of Improvement; **DASS-21** = Depression Anxiety Stress Scale; **EUROHIS-QOL** = Eurohis Quality of Life Scale; **FNE** = Fear of Negative Evaluation Scale; **FNE-B** = Fear of Negative Evaluation Scale—Brief Version; **GPQ** = Gatineau Presence Questionnaire; **ISS** = Internalized Shame Scale; **KSAD** = Korean Social Avoidance and Distress Scale; **LSAS** = Liebowitz Social Anxiety Scale; **LSAS-SR** = Liebowitz Social Anxiety Scale—Self Reported; **M.I.N.I.** = Mini-international neuropsychiatric interview **PRCS** = Personal Report of Confidence as a Speaker; **PDBQ** = Personality Disorder Belief Questionnaire; **PERS** = Post Event Rumination Scale; **PQ** = Presence Questionnaire; **SCID** = Structured Clinical Interview; **SIAS** = Social Interaction Anxiety Scale; **SPRS** = Social Performance Rating Scale; **SPS** = Social Phobia Scale; **SSPS** = Self-statements during public speaking; **SSQ** = Simulator Sickness Questionnaire; **STAI** = State-Trait Anxiety Inventory; **SWEAT** = Specific Work for Exposure Applied in Therapy.

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
