# Peer review of "The Simulation Game—Virtual Reality Therapy for the Treatment of Social Anxiety Disorder: A Systematic Review"

_ijerph, 2021, doi:10.3390/ijerph182413209_

Round 1
Reviewer 1 Report
Corrections and suggestions have been added to the manuscript in this version. The authors entered additional information that contributed to a dense text on the topic.
Author Response
Thanks for your comments
Reviewer 2 Report
I completed my review of the current manuscript and found it improved if compared to the original version. However, I feel that some relevant issues should be further addressed. Therefore, I invite the authors to tackle my concerns listed below.
- English should be revised throughout by a native English speaker.
- Based on my reading, I feel that the authors currently considered both immersive virtual reality and non-immersive virtual reality but it would be prefarble to explicitly clarify that point along eligibiility criteria.
- In the abstract the authors state that a 10-year range interval of publication time was included. Conversely, in the ligibility criteria it was stated that no specific time limits were embedded. Clarification is needed.
- The authors claimed that huge costs represent a limit of the in vivo exposure therapy. Unfortunately, the authors do not argue on the Virtual reality costs. I mean, at least a comparison between both approaches should be detailed. The sustainability of a VR or a iVR-based intervention should be included. Does iVR is really so affordable, cheap, and/or not so expensive? Clarificartion is needed.
Author Response
Dear Reviewer, we thank you for your invaluable and clear advice that has allowed our paper to improve.
In particular:
1. The whole article has been grammatically reviewed by a Cambridge C2 English reviewer, we hope
this has allowed us to eliminate any grammar errors.
2. With regard to point 2, within the article we have considered only the IVR with HMD, however,
after its reporting we realized a gap that can mislead the reader - along paragraph 3.2 in fact, the
use explicit HMD is specified only for the research of Kim et al. - the reason for this error and that
only this research specified the "brand" of the HMD used, which led to specify it in this research
and forgetting in the others, thus giving the reader the feeling that these are incremental levels of
different VR.
We therefore specified for each research that VR therapy was conducted with specific HMDs,
identified as the main tool of IVR in our introduction, and especially how this therapy was
administered (paragraphs 215-221 | 240-247 | 300-302 | 379-382).
3. This point is also the result of a small inattention that we are grateful to have reported to us, the
bibliography search was conducted without time limits, but the articles that met the eligibility
criteria were then shown to be enclosed between 2009 (Wallach et al.) and 2020 (Kim et al.) this
led to erroneous writing that the articles spanned 10 years, but this was not critical and relevant
information with what our eligibility criteria were, therefore the wording on the abstract is been
deleted.
4. Not discussing fully the costs of VR was indeed a lacuna in the text, therefore in the introduction
(paragraphs 108 to 127) an analysis of the sustainability in terms of costs of the VR intervention
was added, citing in particular the research by Robillard et al. , which validate a questionnaire on
the "cost and effort required" analysis, a questionnaire present on one of the researches of the
systematic review. In doing so, therefore, the Robillard analysis cited in the introduction finds
empirical evidence in paragraphs 413-416 which affirm that SWEAT has demonstrated a
“convenience” of VRET also as regards the treatment of SAD.
We really hope that our corrections have satisfied your requests.
Kind regards,
The authors
This manuscript is a resubmission of an earlier submission. The following is a list of the peer review reports and author responses from that submission.
Round 1
Reviewer 1 Report
Dear,
This text does not present any relevant scientific contribution. Furthermore, it does not follow any consistent scientific rigor for the data collection procedure. This is evident when the authors state the following excerpt in the manuscript:
"In our opinion, this research, is the most complete to explain the therapy in Virtual. Reality precisely because it takes into consideration all its facets."
I believe the authors were wrong to submit such an opinion text to this prestigious scientific journal. I strongly recommend rejecting and submitting authors to another journal with adequate scope, comic book maybe.
Author Response
Dear Reviewer,
We are sorry that our systematic review did not meet your needs.
We want to be as rigorous as possible, so even in the face of a negative opinion we took advantage of your report and tried to improve the most critical points
for example, we have eliminated opinions that may seem personal.
thanks for your attention
Reviewer 2 Report
I completed my review of the current submission which carried out a systematic review of the literature on the use of Immersive Virtual Reality for the treatment of Social Anxiety Disorder. I found the manuscript interesting and the topic appropriate for the Journal. However, I feel that major issues should be addressed in a suitable revision. Therefore, I invite the authors to tackle my concerns listed below.
- As minor revisions, acronyms should be spelled out the first time. I suggest to use an identical terminology throughout the paper. For example, subjects and participants on one side and Eligibility/ Inclusion criteria on the other should be modified.
- The Introduction should be improved. A solid theoretical framework on the use of virtual reality for anxiety disorders currently missing should be provided. A strong rationale should be claimed accordingly.
- I'm puzzled in the Method section by reading among eligibility criteria (page 3, lines 115-116), if correctly understood, only positive results were considered? Clarification is mandatory in this regard.
- The Discussion section should be improved. The implications of the findings should be critically argued. Relevant citations should be added. The overall relevance of the paper should be further clarified.
- The Conclusion section should be enhanced. Future research perspectives within this topic/theoretical framework should be stated.
Author Response
Dear Reviewer,
We thank you for taking the time to review our "systematic review", we have tried with this manuscript to correct some "limitations" that have been highlighted.
- As requested, now the acronyms are all the same, by doing so the terminology used should be clearer to the reader (for example as requested, we have eliminated the opposition between "eligibility criteria" and "inclusion criteria" and we have only used "eligibility" / the same was done for the dichotomy participants | subjects)
- We have tried to enrich the introduction, especially in the SAD part, by doing so we have tried to explain which are the most used therapeutic protocols and why we hypothesize to add virtual reality therapy to them. If we have not gone into too much detail on the introduction, it is because, being a systematic review of the literature, it seemed redundant to us to speak specifically in the introduction of the therapies discussed later. however, we hope to have grasped the suggestion well.
- We apologize if the methods section, in particular the "outcomes" have been misunderstood, obviously we have not considered "only" the positive results, on the contrary in the name of the critical spirit that every research must follow we want the results to be as clear as possible at the reader, the fact that not only positive results are considered, among other things, is confirmed by Kampmann's research where iVET is superior to VRET - we have therefore corrected the wording of the outcome to make it clearer to the reader. let's hope it goes well now
- We have tried to treat the "discussion" more critically, adding specific citations and references to the measures and research used. By doing so we hope to have clarified the implications of the findings of the authors covered in the systematic review.
- We have broadened the conclusions, and as requested we have specified where future research in the field of VRET should set their goals. Here too we hope to have correctly grasped the correction that has been placed.
We thank you for your kind attention,
kind regards.
Round 2
Reviewer 1 Report
The theme presented in this manuscript is interesting, it has a good potential for publication since that be extensively modified and improved completely.
- It is recommended that the title has up to 11 words.
- what advantages of the applied methodology in this study in regard to others, (describe on the abstract)
- Introduction should be incremented with more information and citations of recent reports. (doi: 10.7861/fhj.2019-0036) (doi.org/10.3389/frvir.2021.692103) (dx.doi.org/10.1136/bmjopen-2020-046986), and others.
- Figure 1 is unreadable (low resolution). I suggest that the authors improve figure quality to facilitate the visualization of the methodological steps.
- o Text in review boxes must be added in the body of the manuscript.
- Review articles need extensive information and citations from the literature to support the review of the authors and present novel scientific knowledge. I suggest that authors raise the reference quantity. It is expected more than 50, (doi: 10.7861/fhj.2019-0036) (doi.org/10.3389/frvir.2021.692103) (dx.doi.org/10.1136/bmjopen-2020-046986), can help you.
Reviewer 2 Report
I reviewed the revised version of the manuscript which I found improved. However, two substantial issues and one optional minor concern should be further addressed in a suitable revision.
(1) Although the topic was conceptualized, I feel that a solid theoretical background useful to justify the rationale of the current review should be clarified and exhaustively provided. It was unclear to me what has been done up to date on this specific framework and what this paper adds.
(2) The practical implications of the findings in the Discussion section should be outlined and critically argued.
(3) As minor issue, I woould substitute the word "subject" with "person" or "user", and/or "participant" throughoout. Additionally, I would remove academic title of the cited author (i.e., Prof. Giacomo Rizzolatti).